# Tailoring of a visible-light-absorbing biaxial ferroelectric towards broadband self-driven photodetection

Shiguo Han[1,2,3], Maofan Li[1,2], Yi Liu[1,2,3], Wuqian Guo[1,2,3], Mao-Chun Hong[1,2], Zhihua Sun [1,2✉] & Junhua Luo [1,2✉]

In terms of strong light-polarization coupling, ferroelectric materials with bulk photovoltaic effects afford a promising avenue for optoelectronic devices. However, due to severe polarization deterioration caused by leakage current of photoexcited carriers, most of ferroelectrics are merely capable of absorbing 8–20% of visible-light spectra. Ferroelectrics with the narrow bandgap (<2.0 eV) are still scarce, hindering their practical applications. Here, we present a lead-iodide hybrid biaxial ferroelectric, (isopentylammonium)$_2$(ethylammonium)$_2$Pb$_3$I$_{10}$, which shows large spontaneous polarization (~5.2 μC/cm$^2$) and a narrow direct bandgap (~1.80 eV). Particularly, the symmetry breaking of 4/*mmmFmm*2 species results in its biaxial attributes, which has four equivalent polar directions. Accordingly, exceptional in-plane photovoltaic effects are exploited along the crystallographic [001] and [010] axes directions inside the crystallographic *bc*-plane. The coupling between ferroelectricity and photovoltaic effects endows great possibility toward self-driven photodetection. This study sheds light on future optoelectronic device applications.

---

[1] State Key Laboratory of Structural Chemistry, Fujian Institute of Research on the Structure of Matter, Chinese Academy of Sciences, 155 Yangqiao West Road, Fuzhou, Fujian 350002, PR China. [2] Fujian Science & Technology Innovation Laboratory for Optoelectronic Information of China, 155 Yangqiao West Road, Fuzhou, Fujian 350002, PR China. [3] University of Chinese Academy of Sciences, Chinese Academy of Sciences, No. 19 A Yuquan Road, Beijing 100039, PR China. ✉email: sunzhihua@fjirsm.ac.cn; jhluo@fjirsm.ac.cn

Bulk photovoltaic effects (BPVE) of ferroelectric materials is a strong light-polarization coupling with unique physical attributes[1–4], including ultrahigh anomalous photovoltage, polarization-dependent activity, and steady-state photocurrent in a homogeneous media, etc. In contrast to conventional asymmetric systems of Schottky barrier and p-n junction[5], these merits of ferroelectrics are closely related to spontaneous polarization ($P_s$), which creates an internal electric field at least one order of magnitude higher than that in the p-n junction[6]. The strong electric field allows efficient dissociation of photogenerated charge carrier and further leads to large non-zero shift photocurrents[7]. That is, without an external source, a controllable power supply can be expected in BPVE-active ferroelectrics, which offers an avenue for assembling the newly-conceptual self-driven photodetector devices[8,9]. The ferroelectric based self-driven photodetector don't need complicated interface engineering and fabrication process, and spontaneous polarization in ferroelectrics can be modified by electrical field, offering multiple electrically tunable functionalities[5,10]. Currently, the mainstream of ferroelectric oxides still suffer from wide bandgap ($E_g$), such as BaTiO$_3$ (~3.2 eV), LiNbO$_3$ (~3.6 eV), BiFeO$_3$ (~2.7 eV), and Pb(Zr,Ti)O$_3$ (~3.6 eV); this suggests only 8–20% of solar spectrum can be absorbed, greatly hindering their broadband device applications[11]. Although photovoltaic effects in ferroelectrics have been known for more than 60 years, BPVE-active ferroelectrics with the $E_g$ smaller than 2.0 eV remain scarce. In this context, significant endeavors should be focused on exploring the visible-light-absorbing ferroelectric candidates with strong BPVEs, because the current leakage of photo-excited charge carriers will cause severe deterioration of electric polarization.

Two-dimensional (2D) hybrid perovskites with the formula of $(A')_2(A)_{n-1}M_nX_{3n+1}$, where A' and A are organic cations, M is metal and X is halogen, are booming as optoelectronic materials[12]. In contrast to the three-dimensional AMX$_3$ prototype subject to the Goldschmidt's tolerance factor, this 2D branch exhibits infinite structural flexibility and tunability. It should be emphasized that, if the flexible aliphatic organic spacer (A') is adopted, a large degree of molecular freedom for dynamic motions tends to trigger phase transition and favor the generation of ferroelectricity[13,14]. Further, a delicate bandgap engineering can be performed by tailoring thickness of inorganic sheets (the n value) and/or modifying halogen (X), which possibly fulfills the target of narrow bandgap[15,16]. Of special concern is the 2D lead-iodide counterparts that combine robust ferroelectricity and exotic visible-light-absorbing capacity, affording opportunities to explore BPVE-active ferroelectrics. For instance, a few

2D lead-iodide ferroelectrics, such as (4,4-difluorocyclohexylammnium)$_2$PbI$_4$ and [4-(aminomethyl) piperidinium]$_2$PbI$_4$ have been recently reported[17,18]. Nevertheless, owing to the monolayered inorganic perovskite sheets (n = 1), they still possess the relatively wide bandgap ($E_g > 2.3$ eV) along with the lacking of ferroelectric BPVEs. One effective pathway is to increase thickness of inorganic sheets (the n value) while retaining ferroelectricity of 2D lead-iodide hybrid perovskites. Consequently, we attempted to perform structure tailoring to balance the ferroelectricity and optical absorption, which is not only crucial for exploring BPVE-active ferroelectrics but also for deepening the understanding on relationship between electric polarization and BPVEs.

We here present a visible-light-absorbing biaxial ferroelectric in the family of 2D hybrid perovskites, (iso-pentylammonium)$_2$(ethylammonium)$_2$Pb$_3$I$_{10}$ (PEPI), which has a large $P_s$ value of 5.2 μC/cm$^2$ and a narrow direct bandgap ($E_g = 1.8$ eV). Most notably, exceptional in-plane BPVEs are observed along its crystallographic [001] and [010] axes inside the bc-plane, stemming from its biaxial ferroelectricity. Such unique BPVEs allow the broadband self-driven photoactivities of PEPI in the wide range of 365–670 nm, showing the high-density photocurrents (1.5 μA/cm$^2$) and large switching ratio (>10$^5$). This work paves a pathway for the design of photo-ferroelectric materials, and expanding their desirable properties for newly-conceptual optoelectronic device applications.

## Results

High-quality dark red crystals of PEPI were grown by the temperature cooling method (Supplementary Fig. 1), and the phase purity and thermal stabilities have been confirmed by powder X-ray diffraction and thermogravimetric analysis, respectively (Supplementary Fig. 2 and Supplementary Fig. 3). Preliminary differential scanning calorimetry (DSC) measurements display two pairs of the exothermic/endothermic peaks in the cooling/heating modes (Fig. 1a), suggesting the occurrence of successive reversible phase transitions. Our subsequent studies disclose that PEPI undergoes the ferroelectric-to-antiferroelectric phase transition ($T_1 = 313$ K), and antiferroelectric-to-paraelectric phase transition ($T_2 = 340$ K), respectively. For convenience, we label the three phases of PEPI as: ferroelectric phase (FEP, below $T_1$); antiferroelectric phase (AFEP, between $T_1$ and $T_2$), and paraelectric phase (PEP, above $T_2$). Moreover, since dielectric constant ($\varepsilon'$) is closely related to the degree of the electric polarizability, the successive phase transitions in PEPI were solidly confirmed by the noticeable anomalies of its temperature-dependent dielectric constants.

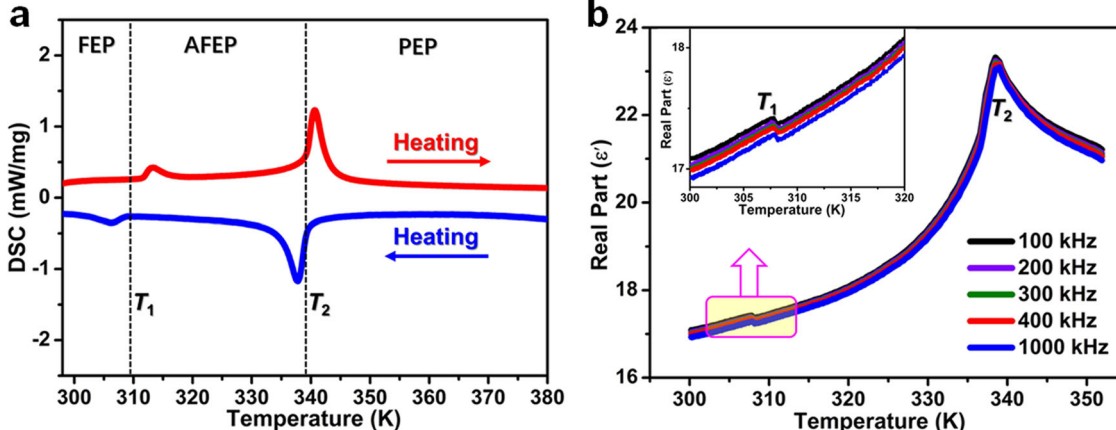

**Fig. 1 Thermal and dielectric properties of PEPI. a** DSC traces and **b** temperature-dependence of the dielectric constants of PEPI performed on single crystals along its $c^{FEP}$-axis direction. Inset: the enlarged view of $\varepsilon'$ at $T_1$.

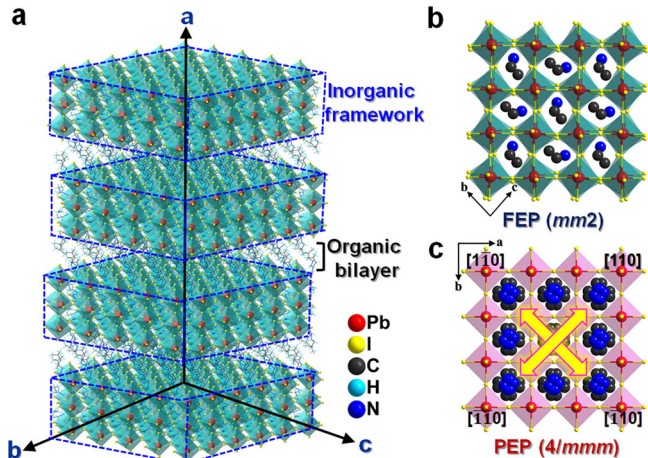

**Fig. 2 Comparison of crystal structures of PEPI in the FEP and PEP. a** Schematic representation of 2D Ruddlesden-Popper type structure for PEPI, with an alternative arrangement of inorganic trilayered perovskite frameworks containing $\{Pb_3I_{10}\}_n$ and organic bilayers of isopentylammonium spacer. **b** Unit cell of PEPI at FEP and (**c**) PEP. The polarization $c$-axis direction at FEP coincides with the [110]-direction at PEP. There are four equivalent [110] directions at PEP, as shown by the green arrowheads.

Figure 1b depicts two different type of dielectric anomalies observed in PEPI; the sharp dielectric peaks at $T_2$ suggest its order-disorder structure change, while the step-like anomaly at $T_1$ should relate to the reorientation of molecular dipoles.

Variable-temperature crystal structures of PEPI were determined by single-crystal X-ray diffraction to deeply probe its phase transition mechanism. At the FEP, it crystallizes in the orthorhombic system with space group $Cmc2_1$ (polar point group $mm2$, Supplementary Table 1). General structure of PEPI adopts a typical 2D Ruddlesden-Popper motif, with an alternative arrangement of inorganic trilayered perovskite frameworks containing $\{Pb_3I_{10}\}_n$ and organic bilayers of iso-pentylammonium spacer. As shown in Fig. 2a, the inorganic sheets are perpendicular to its crystallographic <100> direction, termed as the <100>-oriented family. Notably, organic ethylammonium cation is tightly confined inside the cavity enclosed by the corner-sharing $PbI_6$ octahedra via strong N-H···I hydrogen bonds (Supplementary Fig. 4). All perovskite trilayers of $\{Pb_3I_{10}\}_n$ distribute inside the $bc$-plane and form almost equivalent packings along its [001] and [010] directions, of which the in-plane structural isotropy leads to its biaxial ferroelectricity and in-plane BPVEs. Specially, the incorporation of large-size ethylammonium cation into the perovskite cages suggests severe distortion of $PbI_6$ octahedra, as deduced by the I-Pb-I angles and Pb-I lengths (Supplementary Table 2 and Supplementary Table 3). This distorted geometry of $PbI_6$ octahedra greatly favors the generation of electric polarization. Moreover, organic bilayers of iso-pentylammonium cations are situated in the interlayer space and linked to inorganic sheets by N-H···I bond (Supplementary Fig. 5 and Supplementary Table 4). The head-to-tail arrangement involves that two adjacent layers of spacer cations are staggered to each other with a (1/2 1/2) shift. Molecular dynamics simulation and proton NMR have evidently revealed the fast dynamic motions of organic cations at A and A′ sites afford the driving force to trigger phase transition and ferroelectric polarization, as verified by other 2D ferroelectrics, such as $(BA)_2CsPb_2Br_7$ and $(BA)_2(formamidinium)_2Pb_3Br_{10}$[19,20], etc.

With the temperature increasing between $T_1$ and $T_2$, crystal structure of PEPI belongs to the nonpolar space group of $Pmcn$ (the point group $mmm$). Its perovskite structural motif is still identified as an octahedral tilting architecture, which results in the emergence of local polarization. However, distinct from its ferroelectric state, an antiparallel alignment is assumed to satisfy the requirement of crystallographic symmetry for antiferroelectric order. For instance, both iso-pentylammonium and ethylammonium cations in the unit cell exhibit two orientations, as revealed by the arrowheads in Supplementary Fig. 6. Symmetry-related moieties in the neighboring slab displace in exactly the opposite directions with the same magnitude, canceling out net polarization of the unit cell. Accordingly, the tilting of $PbI_6$ octahedra also occurs, as verified by the off-center displacements of Pb ions. Such a collaboration gives rise to antiparallel dipole arrangement for PEPI, along with the elimination of net electric polarization at AFEP (i.e. $P_s = 0$). Further heating beyond $T_2$, PEPI transforms to the tetragonal system with a nonpolar space group of $I4/mmm$ (the point group of $4/mmm$). The characteristic is that both organic isopentylammonium and ethylammonium cations become highly disordered and locate on the crystallographic mirror plane (Supplementary Fig. 7); all the inorganic $PbI_6$ octahedra feature a highly-symmetric configuration. This centrosymmetric packing totally eliminates the polarization of PEPI, corresponding to its para-electric state. Thus, symmetry breaking of $4/mmmFmm2$ can be deduced at $T_1$, which belongs to one of the 88 species for ferroelectrics and reveals multiaxial nature for PEPI[21]. Concretely, the number of symmetric elements decreases from 16 ($E$, $2C_4$, $C_2$, $2C_2'$, $2C_2''$, $i$, $2S_4$, $\sigma_h$, $2\sigma_v$, $2\sigma_d$) to 4 ($E$, $C_2$, $\sigma_v$, $\sigma_v$), which consistent well with the Landau phase transition theory (Supplementary Fig. 8). That is, the polarized direction of PEPI is confined along crystallographic $c$-axis at FEP, coinciding with the [110]-direction at PEP (Fig. 2b). Due to four equivalent [110]-directions at PEP, four equivalent polarization-directions can be verified for PEPI (Fig. 2c); this multiaxial attribute is also confirmed by the polarization versus electric field ($P$-$E$) hysteresis loops measured along the different directions.

Second harmonic generation (SHG) technique has been developed as an effective tool to detect symmetry breaking in ferroelectrics[22]. Here, variable-temperature SHG properties were studied during the successive phase transitions of PEPI. Figure 3a depicts that PEPI is SHG-active at FEP (below $T_1$); with temperature rising, its SHG signal exhibits a gradual decline and becomes basically unresponsive above the $T_1$. This obvious change of SHG effects behaves as a direct proof to symmetry breaking in PEPI. During its FEP-to-PEP phase transition, the sharp pyroelectric current peak in the vicinity of $T_1$ is reminiscent of electric polarization, and the optimum $P_s$ is estimated to be 5.2 μC/cm$^2$ (Fig. 3b). As the most direct evidence of ferroelectricity, the well-shaped rectangular $P$-$E$ hysteresis loops were recorded at 298 K using the Sawyer-Tower circuit method. At the frequency of 25 Hz, the $P$-$E$ loop affords the saturated $P_s$ value of 5.0 μC/cm$^2$ and the coercive electric field ($E_c$) of 9 kV/cm, respectively (Fig. 3c). These figure-of-merits fall in the range of some other organic-inorganic hybrid ferroelectrics, such as TMCM-MnCl$_3$[23]. Emphatically, such perfect rectangular $P$-$E$ hysteresis loops can be facilely achieved not only along the $b^{FEP}$-axis direction, but also along the crystallographic $c^{FEP}$-axis. As shown in Fig. 3c, the experimental $P$-$E$ hysteresis loops have the almost identical motifs, suggesting the biaxial merit of ferroelectricity in PEPI. That is, its electric polarization can be switched along two axes with four equivalent directions. In terms of our

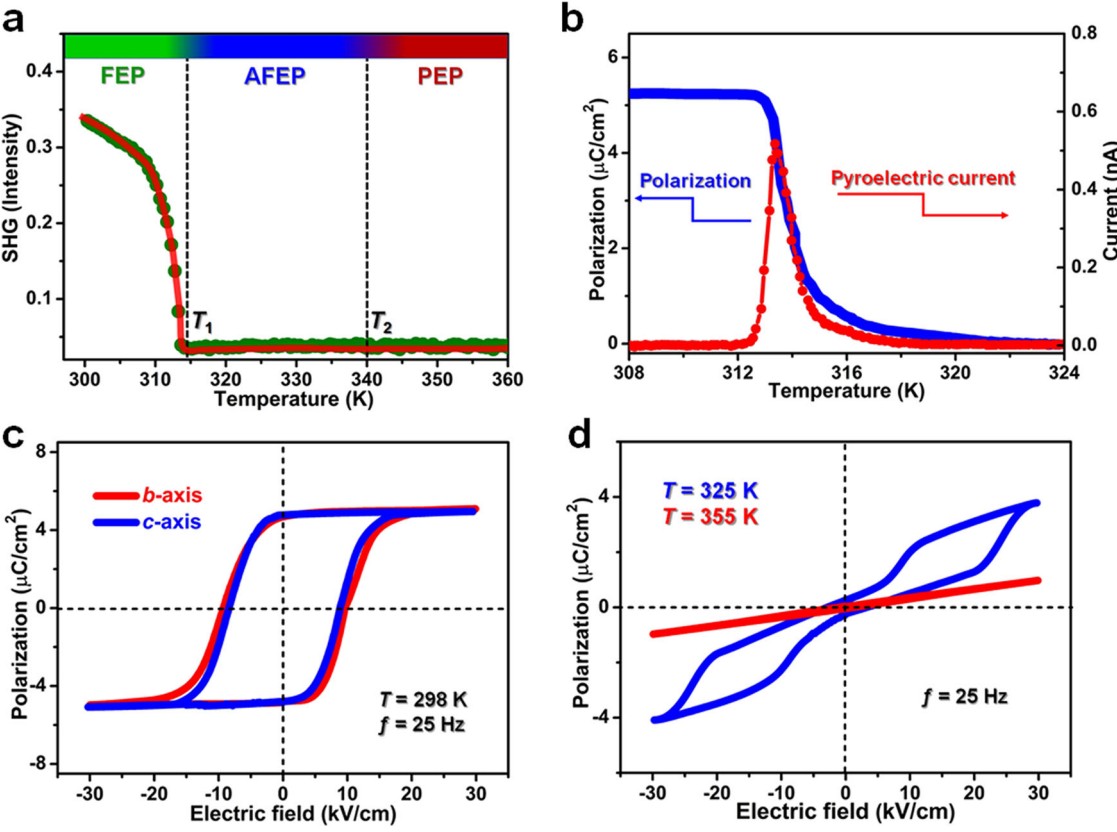

**Fig. 3 Ferroelectric and related properties of PEPI. a** Variable-temperature SHG signals measured in the heating mode. **b** Temperature-dependent $P_s$ deduced from the integral of pyroelectric current. **c** $P$-$E$ hysteresis loops at 298 K. **d** Double $P$-$E$ hysteresis loop collected at 325 K and the linear hysteresis loop at 355 K.

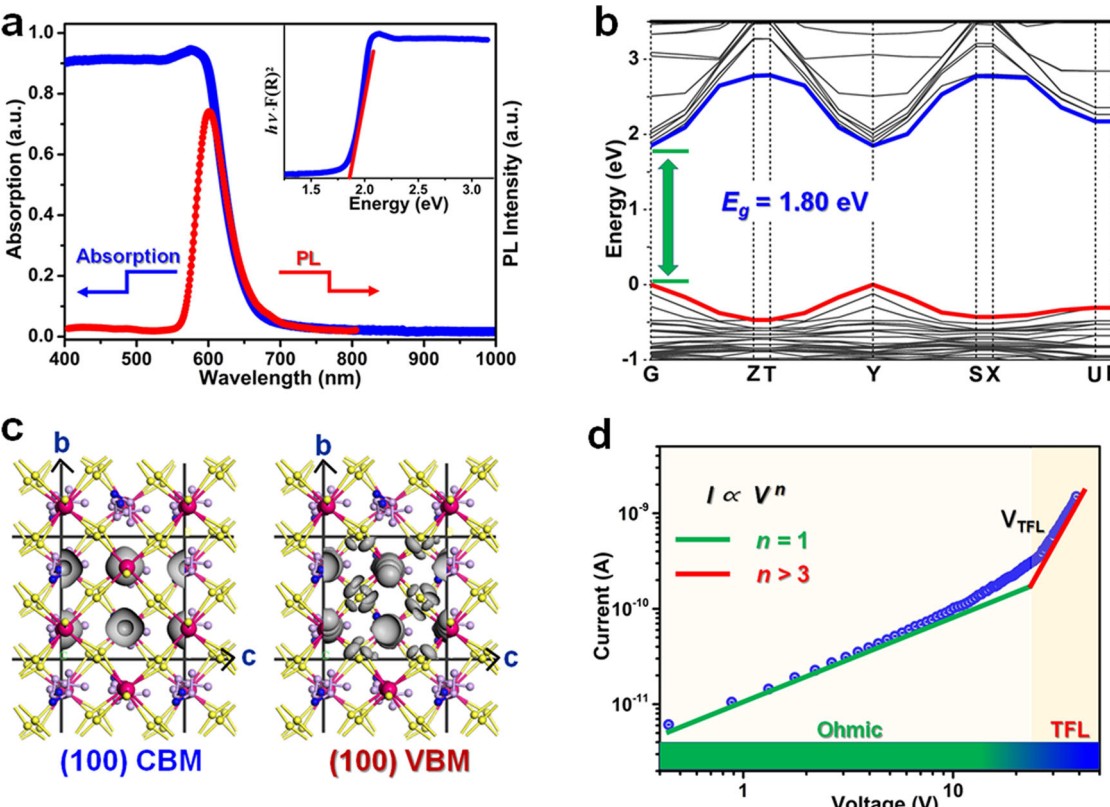

**Fig. 4 Optical and drift mobility characterizations of PEPI. a** Optical absorption and photoluminescence (PL) spectra. Inset: Bandgap calculated from the *Tauc* plot. **b** Calculated band structure. **c** Charge–density distribution of the CBM and VBM inside the crystallographic *bc*-plane. **d** Logarithmic dark $I$–$V$ curve used for the trap-state-density measurement.

structure analyses on its symmetry breaking, it is the in-plane structural isotropy of the crystallographic $bc$-plane that results in equivalent packings along the $b$- and $c$-axes, as well as biaxial ferroelectricity of PEPI. All these findings illuminate that PEPI is a room-temperature biaxial ferroelectric material, which should be a potential candidate for exploiting the in-plane BPVEs. Besides, the antiferroelectric attribute of PEPI is also validated by the double $P$-$E$ hysteresis loop at AFEP (325 K), while the linear straight line discloses its paraelectric feature above the $T_2$ (Fig. 3d).

For ferroelectric materials, it is an intrinsic contradiction between superior excellent ferroelectricity and narrow optical bandgap; that is, leakage current caused by the increasing of photoexcited carriers would result in the deterioration of polarization[24]. The most of oxide ferroelectrics, such as BaTiO₃, P(Zr,Ti)O₃ and BiFeO₃, suffer from their wide bandgap in the range of 2.7~4 eV, which enable the usage of only 8–20% of solar spectrum[11,25,26]. Here, the sharp absorption edge at 670 nm and PL peak at 603 nm are observed for PEPI, of which the $E_g$ is estimated as 1.82 eV based on the *Tauc* equation (Fig. 4a). First-principle calculations on the electronic structure disclose that the conduction band minimum (CBM) and valence band maximum (VBM) of PEPI coherently locate at the $G$ point, indicating its direct-bandgap feature, and the calculated $E_g$ of 1.80 eV coincides with our experimental result (Fig. 4b). As far as we know, PEPI possess a narrow bandgap in molecular ferroelectrics

system, much smaller than those of other lead-iodide ferroelectrics (Supplementary Table 5), such as (4,4-difluorocyclohexylammnium)₂PbI₄ (~2.38 eV)[17], [4-(aminomethyl)-piperidinium]₂PbI₄ (~2.38 eV)[18], [(CH₃)₃NCH₂I]PbI₃ (~2.82 eV), R- and S-1-(4-chlorophenyl)-ethylammonium]₂PbI₄ (~2.34 eV)[27,28], etc. This result well balances the contradiction between ferroelectricity and optical absorption, which endows strong light-polarization coupling and remarkable photoactivities. Further analyses on charge-density distributions in-plane disclose that the charge density of CBM and VBM originates from Pb $s$, Pb $p$ and I $p$ states in the inorganic $bc$-plane sheets (Fig. 4c). The almost isotropy of charge-density distributions in the (100) plane result in its biaxial ferroelectricity and in-plane BPVEs. For the VBM, the strong antibonding coupling of Pb $s$ and I $p$ states favors a small hole effective mass comparable with electron effective mass[29]. Generally, charge transport dynamics of materials are essential for their semiconducting properties. Current-voltage ($I$-$V$) response followed by space charge limited current analysis was measured to validate the transport properties of crystal PEPI. Figure 4d shows that the logarithmic $I$-$V$ trace has two different regimes: ohmic region ($n = 1$) and trap-filled limited (TFL) regime ($n > 3$). The drift mobility is extracted to be 0.5 cm² V⁻¹s⁻¹, much larger than that of other ferroelectrics, indicating excellent semiconductor properties of PEPI. Moreover, Ultraviolet photoelectron spectroscopy measurement shows that the Fermi level of PEPI deviate from the middle of the bandgap and is close to the valence band, which indicates that PEPI is $P$-type semiconductor material (Supplementary Fig. 9)[30].

The notable biaxial ferroelectricity and excellent semiconducting feature along the (100) plane of PEPI offer great opportunities to obtain the in-plane BPVEs. Short-circuit photocurrent ($I_{sc}$) were measured on surface of PEPI under incident illumination of 637 nm without the bias voltage applied (Fig. 5a). More specifically, in order to explore the polarization-induced BPVEs, lateral-type gold electrodes were fabricated aligned parallel/perpendicular to the direction of (100) plane. As shown in Fig. 5b, for the electrodes aligning in the (100) plane (A1 and A2), prominent $I_{sc}$ (~6 nA) were observed under the incident illumination intensity of 30 mW/cm². However, for the electrodes aligning perpendicular to the direction of (100) plane (A3), no $I_{sc}$ signal could be detected. That is, exceptional in-plane ferroelectric BPVEs are exploited in the (100) plane. As far as we are aware, the ferroelectric polarization-induced electric field of (100) plane plays an important role in such interesting phenomenon. From another point of view, the evident symmetry breaking with 4/ $mmm$F$mm$2 species of PEPI result in two equivalent polar axes ($b$-axis and $c$-axis). Then, the in-plane BPVEs of PEPI are closely related to its ferroelectric polarization-induced electric field of (100) plane.

In principle, ferroelectrics with BPVE cloud offer the power supply for itself. Based on such excellent in-plane BPVE, PEPI could behave as a promising candidate for broadband polarization self-driven photodetection. As shown in Fig. 6a, $I$–$V$ measurements were performed along the $b$-, $c$- and $a$-axis, respectively. The obvious polarization self-driven effects are easily obtained along the $b$-, $c$-axis, including zero-bias $I_{sc}$ and open circuit voltage ($V_{oc}$). However, there is no $I_{sc}$ and $V_{oc}$ signal along the $a$-axis (Insert of Fig. 6a). That is, self-driven photodetection behaviors should be only exploited along the $b$- and $c$-axis direction. This behavior is well consistent with the in-plane (100) BPVE of PEPI. Figure 6b presents the photocurrent density dependence of voltage bias of PEPI illuminated with a 637 nm laser along the polar axis direction. Specifically, under illumination of 127 mW/cm², the zero-bias $I_{sc}$ and $V_{oc}$ measured to be ~1.5 µA/cm² and ~0.8 V, respectively. Such short circuit current is

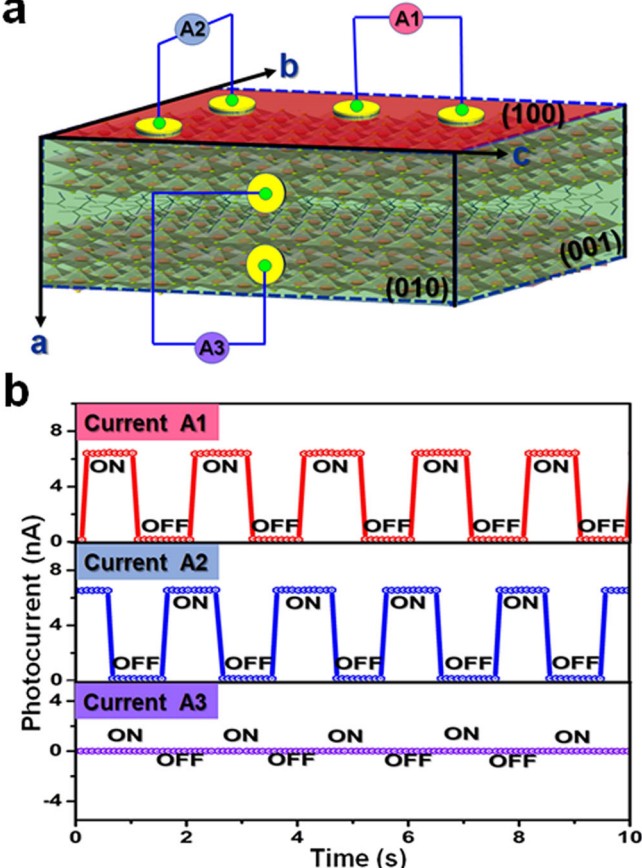

**Fig. 5 Transient photocurrent response of PEPI along different crystallographic directions. a** Schematic diagram for in-plane ferroelectric BPVEs measurements with electrodes aligned parallel (A1, A2) and perpendicular (A3) to the direction of (100) plane. **b** Short-circuit photocurrent measurements in the light "on/off" cycles.

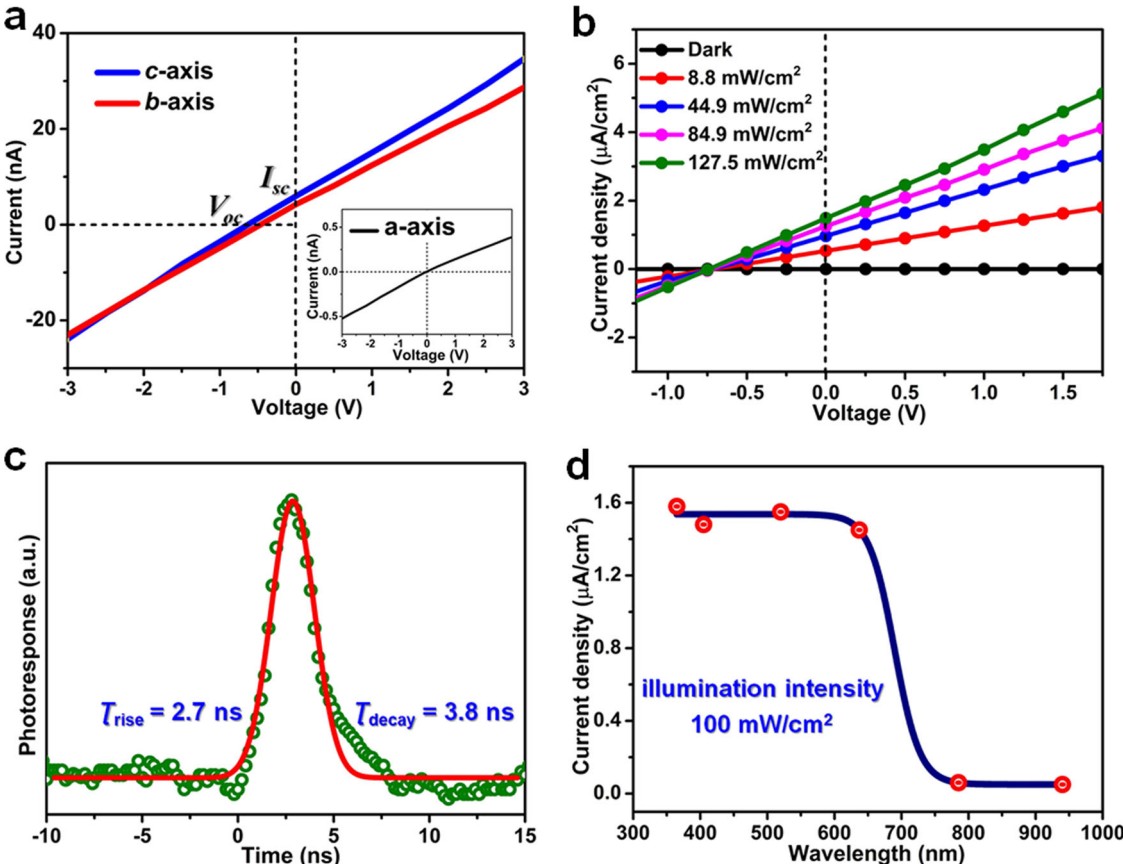

**Fig. 6 Self-driven photodetection properties of PEPI. a** I–V measurements of PEPI measured along the b-, c-, and a-axis, respectively. **b** Photocurrent density dependence of voltage bias of PEPI illuminated with a 637 nm laser. **c** Transient photoelectric responses under a nanosecond 355 nm laser. **d** Short-circuit photocurrent as a function of illumination wavelength at the zero bias.

slightly larger than the most active ferroelectric oxide BiFeO₃ (~0.4 μA/cm²)[31], and superior to those of some ferroelectric oxides, such as $[KNbO_3]_{1-x}[BaNi_{1/2}Nb_{1/2}O_{3-\delta}]x$ (~0.1 μA/cm²) under the halogen lamp (~4 mW/cm²), (Pb, La)(Zr,Ti)O₃ (8 nA/cm²) and (Na, K)NbO₃ (~25 nA/cm²) under ultraviolet illumination[32,33]. In addition, the dark current ($I_{dark}$ ~1 × 10⁻¹⁴ A) is lower at least by 2 orders than reported traditional p-n based devices[34]. Such extremely low $I_{dark}$ and considerable $I_{sc}$ generate a large switching ratio of 10⁵ at the zero bias, which indicates the excellent self-driven photodetection properties of PEPI. Besides, the I-V curves of the crystal detector after 7 days exhibit slight difference compared with the fresh device disclose that PEPI possesses fairly good phase stabilities (Supplementary Fig. 10). More interestingly, the transient response speed of PEPI can reach to the nanosecond level (~2.7 and ~3.8 ns) under a nanosecond laser illumination (Fig. 6c). Such ultrafast response reveals the rapid carrier recombination process and low electron-hole pair transit time, which could broaden the application scope of photoelectric devices, including high-frequency optical communication, ultrafast dynamic process simulation et al. Moreover, the wavelength-dependent $I_{sc}$ of PEPI was carried out ranging from 365 to 940 nm. As shown in Fig. 6d, PEPI exhibits obvious $I_{sc}$ signal in the wide range of 365–670 nm, which is consistent with its narrow optical absorption bandgap. Compared to traditional inorganic materials, such as ZnO[9], BiFeO₃[35], Pb(Zr,Ti)O₃[36], and GaN[37], PEPI possesses the wider self-driven detection range up to 670 nm. As far as we know, this figure-of-merit is the widest value obtained for the existing molecular ferroelectrics materials, which endows great potentials of PEPI toward broadband self-driven photodetection.

## Discussion

Particularly, with the temperature increasing, the circuit voltage decreases gradually and disappears completely above the $T_1$ (Fig. 7a). This tendency coincides with that of the temperature-dependent polarization (Fig. 3b), being reminiscent of the close relationship with its ferroelectricity. We further studied the switching of photovoltage after application of 1500 V voltage pulses to the device. It is obvious that both the direction of $V_{oc}$ and $I_{sc}$ can be reversed by inverting electric poling, while the magnitude keeps almost unchanged (Fig. 7b). Such a flipping of photo-induced voltage and current attribute to the direction change of electric polarization in PEPI, which has been verified in ferroelectrics of BiFeO₃ and PLZT ceramics[32]. The origin of BPVE in ferroelectrics has been extensively studied and some speculative models were also suggested on inorganic materials, such as the depolarization-dependent band bending at metal-ferroelectric interface, asymmetry potentials related to the depolarization field, and nonlinear optical-rectification effect, etc[38–41]. As far as we are aware, all these scenarios state that polarization makes a critical role in BPVE. In this regard, the sign of photocurrent and photovoltage can be reversed by inverting poling, which involves the switching of polarization. Here, it is supposed that photocurrent switching of PEPI also involves with the separation and flipping of polarization-related photoexcited carriers (Fig. 7c). For PEPI, the dynamic motions of organic moieties are restrained with temperature lowering below the $T_1$. From a structural point of view, this preferred

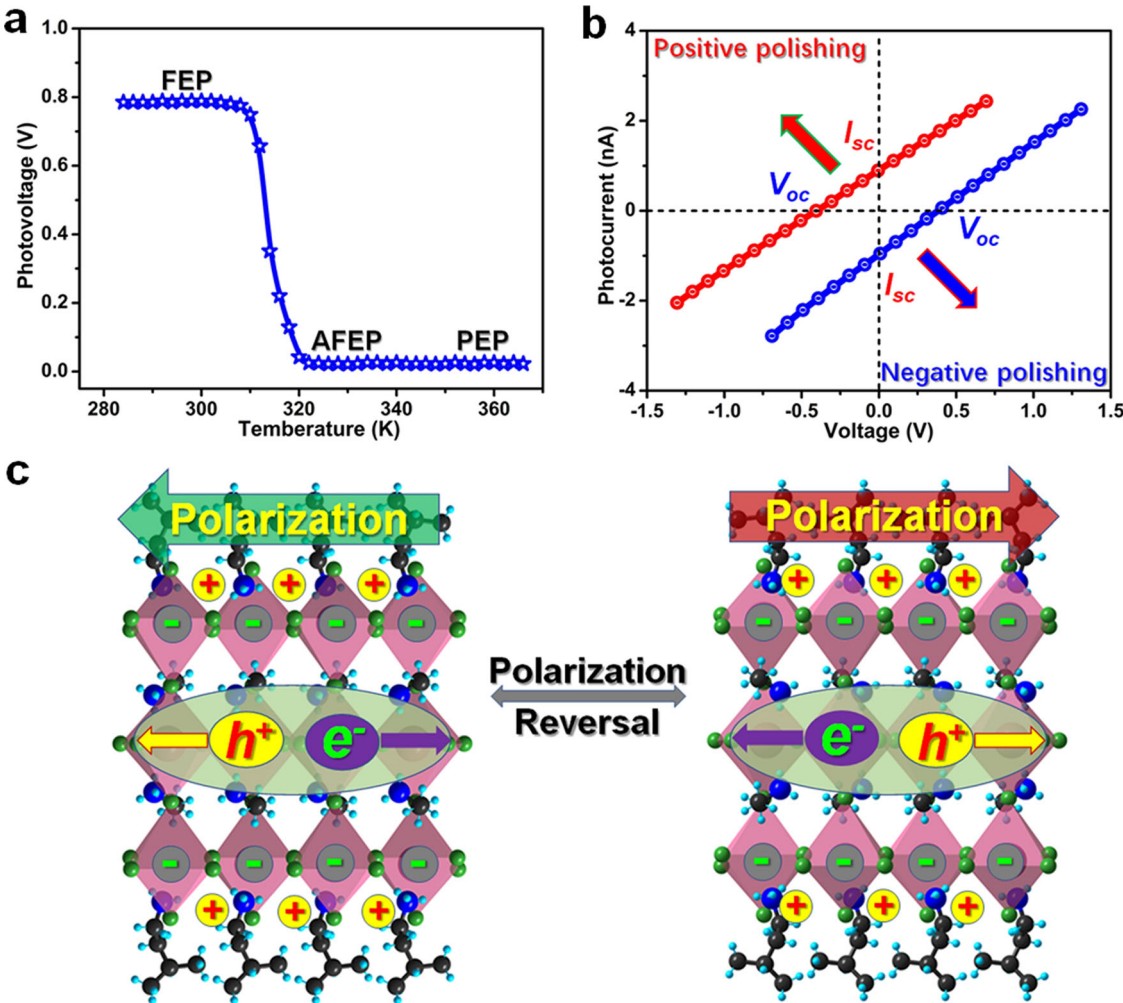

**Fig. 7 Polarization switching and related properties of PEPI. a** Photovoltage dependence of temperature for PEPI. **b** $I–V$ characteristics with the inversion of polarized direction. Both the direction of $V_{oc}$ and $I_{sc}$ can be reversed by inverting electric poling. **c** Schematic illustration of switchable photocurrent directions with polarization-related photoexcited carriers with the charge separation under positive polarization direction and negative polarization direction.

orientation of organic cations results in the breaking of its prototypic mirror symmetry, and thus creates a degree of long-range dipolar order. Besides, the inorganic framework of PEPI is also strongly influenced by the molecular orientation. Therefore, the electric switching and temperature-dependence of the BPVE behavior confirms the role of ferroelectricity in PEPI, which stems from the dynamic reorientation of organic cations and tilting motions of inorganic $PbI_6$ octahedra.

In summary, we have successfully reported a visible-light-absorbing molecular ferroelectric: (isopentylammonium)$_2$ (ethylammonium)$_2Pb_3I_{10}$ (PEPI), which possesses a narrow bandgap ~1.80 eV. Emphatically, PEPI exhibits exceptional in-plane BPVEs in the (100) plane, stemming from its bi-axial ferroelectricity. Such BPVE activities and narrow bandgap endow intriguing self-powered photoactivities in a broadband range of 365–670 nm, which far more superior than conventional perovskite structural metal oxides. Without external energy source, PEPI creates switchable photovoltaic current with high current density (~1.5 μA/cm$^2$) and large switching ratio (>10$^5$), beyond the most active oxide counterpart of $BiFeO_3$. Structural analyses reveal that this unique self-driven photodetection behavior of PEPI stem from dynamic ordering of organic cations, along with

the tilting motion of inorganic $PbI_6$ octahedra. Our work allows for a deep understanding of the origin of ferroelectric self-powered photodetection effects, and will undoubtedly promote their further optoelectronic application potentials.

## Methods

**Materials**. Lead acetate trihydrate (Pb(Ac)$_2$, 99.5%, Aladdin), isopentylamine (99%, Aladdin), ethylamine (68–72% in $H_2O$), hydroiodic acid (HI, 48%, Aladdin). All the chemicals were bought and used without further purification.

**Synthesis**. Compound PEPI was synthesized from the concentrated aqueous HI solutions (57%). Firstly, Pb(Ac)$_2$ (5.0 g) was dissolved in 48% aqueous HI solution (25 mL) by heating to boil under constant magnetic stirring to give a yellowish solution. Subsequent addition of isopentylamine (0.4 g) and ethylamine (0.75 g) to the hot solution formed dark red precipitation. Then, the reaction mixture was stirred with heating for 30 min to dissolve the components thoroughly. Finally, large-size crystals were grown by the temperature cooling method with the temperature lowing rate of 1 K/day. The phase purity was verified by the infrared spectrum (Supplementary Fig. 11), $^1$H-NMR (Supplementary Fig. 12), and $^{13}$C-NMR (Supplementary Fig. 13).

**Thermal measurements**. Differential scanning calorimetry (DSC) measurement was performed using the NETZSCH DSC 200 F3 with the heating/cooling rates of 5 K/min. Thermogravimetric-differential scanning calorimetry (TG-DSC) analysis was carried out on a Netzsch STA 449 C unit.

**Single-crystal X-ray diffraction**. Variable-temperature single-crystal diffraction data of PEPI were collected on a Bruker D8 Quest/Venture diffractometer with Mo–Ka radiation ($\lambda = 0.77$ Å) at 260 K, 320 K, and 350 K, respectively.

**Optical measurements**. Ultraviolet-visible diffuse reflectance spectrometry was performed on a PerkinElmer Lambda 950 UV-vis-IR spectrophotometer.

**Electrical measurements**. The dielectric constants were measured using the two-probe AC impedance method with an Impedance Analyzer (TH2828A). Photoelectric and pyroelectric measurements were measured using a Keithley 6517B source meter. The response speed of our detector was measured by the transient photocurrent method, using the 355 nm pulse laser with a width of 3 ns. High-speed oscilloscope (Tektronix MDO3014) with the frequency of 1 GHz to record response signals.

**Electronic properties**. Ultraviolet photoelectron spectroscopy (UPS) measurement was conducted on a Thermo Scientific ESCALAB 250Xi XPS system.

**Ferroelectric measurements**. Single-crystal sample with a thickness of ~1 mm was cut and polished for the measurements of ferroelectric properties. The polarization versus electric field ($P$–$E$) hysteresis loops were measured on a ferroelectric analyzer (Radiant Precision Premier II) using the Sawyer-Tower circuit method.

**Computational methods**. Density function theory (DFT) calculations were performed using the plane-wave pseudopotential method implemented in the Cambridge Sequential Total Energy Package (CASTEP) software package.

## Data availability

The authors declare that all data supplementary to the findings of this study are available within the paper and its supplementary information files. The structures have been deposited at the Cambridge Crystallographic Data Centre (deposition numbers: CCDC 1950869, 1950872 and 1950875), and can be obtained free of charge from the CCDC via www.ccdc.cam.ac.uk/getstructures. Any further relevant data are available from the authors upon reasonable request.

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

## Acknowledgements

This work was supported by the National Natural Science Foundation of China (21875251, 21833010, 21525104, and 21921001), the Key Research Program of Frontier Sciences of the Chinese Academy of Sciences (ZDBS-LY-SLH024), the NSF of Fujian Province (2018H0047), the Strategic Priority Research Program of the CAS (XDB20010200), and Youth Innovation Promotion of CAS (2019301 and 2020307).

## Author contributions

S.G.H. prepared the samples and wrote the manuscript. M.F.L. and Y.L. determined the structures. W.Q.G. measured the ultraviolet-visible absorption spectroscopy. M.C.H. provided suggestions for research. Z.H.S. and J.H.L. designed and directed the studies. All authors contributed to write and review the manuscript.

## Competing interests

The authors declare no competing interests.
