## [Peer Review File · Nature Communications]

Reviewer #1 (Remarks to the Author):

The manuscript is on ferroelectricity and in-plane bulk photovoltaic effects (BPVEs) in 2D lead-iodide hybrid perovskite $[(\text{CH}_3)_2\text{CH}(\text{CH}_2)_2\text{NH}_3]_2[(\text{CH}_3\text{CH}_2\text{NH}_3)_2\text{Pb}_3\text{I}_{10}]$. The coupling between ferroelectricity and visible-light-driving in-plane BPVEs is rare. Therefore these preliminary results are interesting for the applied physics community. However, the clarity of the manuscript needs to be improved as follows:

1. The authors confirm that the drift mobility is about $0.5 \text{ cm}^2 \text{ V}^{-1} \text{ s}^{-1}$ in the molecular crystal. Is the semiconductor P-type or n-type? What is the mobility of its main carrier?
2. As the authors said that the photovoltage can be switched by the direction of polarization by inverting electric poling. How to exclude the effect of charge injection or charge trapping after the electric poling. And, does this switching effect maintain for a long time, how long?

Reviewer #2 (Remarks to the Author):

Organic-inorganic hybrid perovskites have emerged as a revolutionary class of electroactive materials in the photoelectric and photovoltaic fields. However, to date, the visible-light-absorbing ferroelectric materials are extremely scarce. In this work, the authors report a new visible-light-absorbing molecular ferroelectric of $(\text{isopentylammonium})_2(\text{ethylammonium})_2\text{Pb}_3\text{I}_{10}$, which possesses the narrowest optical bandgap of molecular ferroelectric systems. It is remarkable that excellent self-powered photoactivities have been demonstrated in a broadband range of 365-670 nm, which are superior than conventional perovskite oxides. In my opinion, this work emphasizes that the importance of hybrid perovskite in photo-ferroelectric and sheds light on their potentials for future photonic device application. Such interesting findings will attract broad interests and wide readerships in the field of material science, chemistry, physics and electronics. Therefore, I strongly recommend the publication of this article in Nature Communications after minor revisions.

1. How the author evaluates thermal stability of this compound, which is quite essential to the subsequent practical applications. It is recommended that thermogravimetric and differential thermal analysis would be a clear indicator in this respect.
2. Compound 1 undergoes obvious symmetry breaking phase transitions from paraelectric phase to ferroelectric phase. The symmetry (elements) variation between its ferroelectric and paraelectric states should be elucidated clearly.
3. Dielectric, ferroelectric and photodetection measurements were performed on crystal samples. If the samples are single crystal samples, which direction is tested? How to grow large crystals in the stable solutions? The detailed process should be provided in supporting information for others to reproduce the crystal growth.

Reviewer #3 (Remarks to the Author):

Shiguo et al., reports photon detection with a ferroelectric material with a bandgap smaller than previously reported material. In principle, it is claimed self-driven can be achieved. It should be of interest for Nature communications. However, to be published in Nature communications, the following comments need to be addressed:

1. There are many other materials including perovskite used as photon detectors. The motivation

using ferroelectric material needs to be clarified. For self-driven, with photovoltaic effect, photons will generate voltage/current. In this case, any such material can be said with self-powering function? It is not well described in the main text. The material used has phase change with different temperature. Probably such device can be used for temperature sensing?

2. Application for Chiral-optics with chiral ferroelectric material is a motivation in this field (See Guankui Long et al., Chiral-perovskite optoelectronics), Will the material used in the current draft be useful for making chiral ferroelectric and be used as circular-polarization photon detectors? These should be discussed.

3. The use of "1" as the presentation of the material reads very strange, this should be replaced with other words.

4. The detection speed seems very fast, but seems beyond authors's instrument detection limit. It is good to see the exact response time for such a fast detector.

5. Stability is normally a problem for such detectors. Is there any such test?

COMMENTS TO AUTHOR:

Reviewer #1:

The manuscript is on ferroelectricity and in-plane bulk photovoltaic effects (BPVEs) in 2D lead-iodide hybrid perovskite $[(\text{CH}_3)_2\text{CH}(\text{CH}_2)_2\text{NH}_3]_2(\text{CH}_3\text{CH}_2\text{NH}_3)_2\text{Pb}_3\text{I}_{10}$. The coupling between ferroelectricity and visible-light-driving in-plane BPVEs is rare. Therefore, these preliminary results are interesting for the applied physics community. However, the clarity of the manuscript needs to be improved as follows:

Q1. The author confirms that the drift mobility is about $0.5 \text{ cm}^2\text{V}^{-1}\text{s}^{-1}$ in the molecular crystal. Is the semiconductor *P*-type or *N*-type? What is the mobility of its main carrier?

Answer: Great thanks for the reviewer's positive comments on our manuscript. Ultraviolet photoelectron spectroscopy (UPS) measurements have been performed to explore the semiconductor properties of our compound (abbreviated as **PEPI**, based on the 3rd reviewer's suggestion). As shown in the following **Figure 1**, UPS result shows that the Fermi level of **PEPI** deviates from the middle of the bandgap and is close to the valence band, indicating that **PEPI** should be the *P*-type semiconductor.¹⁻⁴ That is, hole transport is the main carrier in this semiconductor material. Such results resemble some other inorganic-organic hybrid perovskite, such as $\text{CH}_3\text{NH}_3\text{PbI}_3$ single crystal.⁵

Figure 1. (a) The UPS measurement of **PEPI**. (b) Schematic evolution of Fermi level position and bandgap edge derived from UPS and density function theory (DFT) results. E_{VAC} is vacuum level; E_C is conduction band level; E_F is Fermi level and E_V is valence band level.

References

- [1] Zhou, W. et al. Recent developments of carbon-based electrocatalysts for hydrogen evolution reaction. *Nano. Energy* **28**, 29-43 (2016).
- [2] Prathapani S. et al. Electronic band structure and carrier concentration of formamidinium-cesium mixed cation lead mixed halide hybrid perovskites. *Appl. Phys. Lett.* **112**, 092104 (2018).
- [3] Li, Y. et al. Coordination assembly of 2D ordered organic metal chalcogenides with widely tunable electronic band gaps. *Nat. Commun.* **11**, 261 (2020).
- [4] Leblanc, A. et al. Enhanced Stability and Band Gap Tuning of α -[HC(NH₂)₂]₂PbI₃ Hybrid Perovskite by Large Cation Integration. *ACS Appl. Mater. Interfaces* **11**, 20743-20751 (2019)
- [5] Dong, Q. et al. Electron-hole diffusion lengths > 175 nm in solution-grown CH₃NH₃PbI₃ single crystals. *Science*. **347**, 967-970 (2015).

Q2. As the authors said that the photovoltage can be switched by the direction of polarization by inverting electric poling. How to exclude the effect of charge injection or charge trapping after the electric poling. And, does this switching effect maintain for a long time, how long?

Answer: Thanks a lot for the reviewer's positive comment on our manuscript. Our studies reveal that the photovoltage of **PEPI** can be reversed by inverting electric poling, which attribute to the direction change of its electric polarization. The similar results have been observed in ferroelectrics of BiFeO₃ and PLZT ceramics.⁶ According to the reviewer's suggestion, we have consulted some relevant references about the charge injection and charge trapping. It is the truth that electric field induced charge injection (followed by charge trapping) has impact on ferroelectric polarization switching. For instance, the polarization switching-induced charge injection and subsequent charge trapping at the metal/ferroelectric interface could lead to a negative differential resistance.⁷ According to the literature, most of the research on charge injection or charge trapping is based on ferroelectric thin film samples.⁸⁻¹⁰ For our block samples, the effect

of charge injection and charge trapping may be very small, which still needs further investigations.

Moreover, according to the reviewer's suggestion, we have measured the retention time of photovoltage after polarization switching. Firstly, current-voltage curve was measured after negative polishing (as shown in the following **Figure 2**, the blue line curve A). The curve A exhibits a photovoltage about +0.370 V. Secondly, a positive voltage pulses was applied to the device in order to obtain polarization inversion. Then, the current-voltage curve was measured again (the red line curve B), and the curve B shows a photovoltage about -0.368 V. Two hours later, the current-voltage curve C (the green line) exhibits a photovoltage about - 0.255 V; Four hours later, the current-voltage curve D (the yellow line) exhibits a photovoltage about - 0.174 V; Six hours later, the current-voltage curve E (the gray line) exhibits a photovoltage about - 0.054 V. Eight hours later, the photovoltage of the device decrease gradually and reaches a saturation of -0.04 V. As far as we know, the long retention time of photovoltage also indicates the small effect of charge injection or charge trapping from another perspective.

Figure 2. Time-dependent current-voltage curves of PEPI photodetector illuminated with a 405 nm laser of 204 mW/cm². A represents current-voltage curve after negative polishing and B represents current-voltage curve after negative polishing.

References

- [6] Zhang, J. et al. Enlarging Photovoltaic Effect: Combination of Classic Photoelectric and Ferroelectric Photovoltaic Effects. *Sci. Rep.* **3**, 2109 (2013).
- [7] Li, P. L. et al. An Unusual Mechanism for Negative Differential Resistance in Ferroelectric Nanocapacitors: Polarization Switching-Induced Charge Injection Followed by Charge Trapping. *ACS Appl. Mater. Interfaces* **9**, 27120-27126 (2017).
- [8] Lee, D. et al. Polarity control of carrier injection at ferroelectric/metal interfaces for electrically switchable diode and photovoltaic effects. *PHYSICAL REVIEW B* **84**, 125305 (2011).
- [9] Ji, H. et al. Improvement of Charge Injection Using Ferroelectric Si:HfO₂ As Blocking Layer in MONOS Charge Trapping Memory. *IEEE Journal of the Electron Devices Society* **6**, 121-125 (2018).
- [10] Fan, Z. et al. Ferroelectric Diodes with Charge Injection and Trapping. *PHYSICAL REVIEW APPLIED* **7**, 014020 (2017).

Reviewer #2:

Organic-inorganic hybrid perovskites have emerged as a revolutionary class of electroactive materials in the photoelectric and photovoltaic fields. However, to date, the visible-light-absorbing ferroelectric materials are extremely scarce. In this work, the authors report a new visible-light-absorbing molecular ferroelectric of (isopentylammonium)₂(ethylammonium)₂Pb₃I₁₀, which possesses the narrowest optical bandgap of molecular ferroelectric systems. It is remarkable that excellent self-powered photoactivities have been demonstrated in a broadband range of 365-670 nm, which are superior than conventional perovskite oxides. In my opinion, this work emphasizes that the importance of hybrid perovskite in photo-ferroelectric and sheds light on their potentials for future photonic device application. Such interesting findings will attract

broad interests and wide readerships in the field of material science, chemistry, physics and electronics. Therefore, I strongly recommend the publication of this article in *Nature Communications* after minor revisions.

Q1. How the author evaluates thermal stability of this compound, which is quite essential to the subsequent practical applications. It is recommended that thermogravimetric and differential thermal analysis would be a clear indicator in this respect.

Answer: Great thanks for the reviewer's positive comments on our manuscript. According to the suggestion, thermogravimetric-differential scanning calorimetry (TG-DSC) analysis was carried out on a Netzsch STA 449C unit within the temperature of 300-1100 K with a heating rate of 10 K min⁻¹. As shown in the following **Figure 3**, the TG-DSC curve indicates that **PEPI** can thermally stable up to 560 K without any thermal decomposition (Please see Supporting Information in revision).

Figure 3. The TG-DSC curves of **PEPI** with heating rate 10 K/min.

Q2. Compound **1** undergoes obvious symmetry breaking phase transitions from paraelectric phase to ferroelectric phase. The symmetry (elements) variation between its ferroelectric and paraelectric states should be elucidated clearly.

√**Answer:** Thanks a lot for the reviewer's useful suggestion. According to the suggestion, we have added more details about the symmetry elements variation in the revision. As shown in the following **Figure 4**, the symmetry breaking in **PEPI** meets the requirement of the Aizu notation $4/mmmFmm2$, belonging to the 88 potential ferroelectric phase transitions. Concretely, the number of symmetric elements decreases from 16 ($E, 2C_4, C_2, 2C_2', 2C_2'', i, 2S_4, \sigma_h, 2\sigma_v, 2\sigma_d$) to 4 ($E, C_2, \sigma_v, \sigma_v$), which consistent well with the Landau phase transition theory (Please see Supporting Information in revision).

Figure 4. Symmetry transformation occurs of **PEPI** from the paraelectric phase to the ferroelectric phase. The number of symmetric elements decreases by three quarters from 16 ($E, 2C_4, C_2, 2C_2', 2C_2'', i, 2S_4, \sigma_h, 2\sigma_v, 2\sigma_d$) to 4 ($E, C_2, \sigma_v, \sigma_v$).

Q3. Dielectric, ferroelectric and photodetection measurements were performed on crystal samples. If the samples are single crystal samples, which direction is tested? How to grow large crystals in the stable solutions? The detailed process should be provided in supporting information for others to reproduce the crystal growth.

√**Answer:** Thanks a lot for the reviewer's useful suggestion.

(1) The symmetry breaking of $4/mmmFmm2$ species for **PEPI** results in its biaxial attributes. That is, **PEPI** has four equivalent P_s directions, and the polarization can be switched with an electric field parallel to the crystallographic b - and c -axes. Hence, perfect rectangular polarization *versus* electric field (P - E) hysteresis loops can be facily achieved along the crystallographic b^{FEP} -axis and c^{FEP} -axis direction (Please

see **Figure 3c** in revision). For photodetection measurement, current-voltage (I - V) measurements were performed along the b -, c - and a -axis, respectively. As a result, the obvious zero-bias short-circuit photocurrent (I_{sc}) and open circuit voltage (V_{oc}) are easily obtained along the b -, c -axis. However, there is no I_{sc} and V_{oc} signal along the a -axis (Please see **Figure 6a** in revision). This behavior is well consistent with the in-plane (100) photovoltaic effects of **PEPI**.

(2) According to the suggestion, we have added more details about the growth of crystal, as follows (Please see Supporting Information in revision).

Materials: Lead acetate trihydrate ($\text{Pb}(\text{Ac})_2$, 99.5%, Aladdin), isopentylamine (99%, Aladdin), ethylamine (68%-72% in H_2O), hydroiodic acid (HI, 48%, Aladdin). All the chemicals were bought and used without further purification.

Synthesis: Compound **PEPI** was synthesized from the concentrated aqueous HI solutions (57%). Firstly, $\text{Pb}(\text{Ac})_2$ (5.0 g) was dissolved in 48% aqueous HI solution (25 mL) by heating to boil under constant magnetic stirring to give a yellowish solution. Subsequent addition of isopentylamine (0.4 g) and ethylamine (0.75 g) to the hot solution formed dark red precipitation. Then, the reaction mixture was stirred with heating for 30 min to dissolve the components thoroughly. Finally, large-size crystals were grown by the temperature cooling method with the temperature lowering rate of 1 K/day.

Reviewer #3:

Shiguo *et al.*, reports photon detection with a ferroelectric material with a bandgap smaller than previously reported material. In principle, it is claimed self-driven can be achieved. It should be of interest for *Nature communications*. However, to be published in *Nature communications*, the following comments need to be addressed:

Q1. There are many other materials including perovskite used as photon detectors. The motivation using ferroelectric material needs to be clarified. For self-driven, with photovoltaic effect, photons will generate voltage/current. In this case, any such material can be said with self-powering function? It is not well described in the main text. The

material used has phase change with different temperature. Probably such device can be used for temperature sensing?

√**Answer:** Thanks a lot for the reviewer's positive comment on our manuscript. According to the suggestion, we have added more details about the self-driven photodetection (Please see manuscript in revision). Self-powered photodetection, refers to sensing light signals without an external driving force to separate the photoexcited electron-hole pairs. Traditional self-powered photodetection systems mainly consist of heterojunction and Schottky barriers. Ferroelectrics, capable of instinctive spontaneous polarization, are holding a promise for next-generation self-powered photodetectors. First of all, the large photovoltage achieved in ferroelectrics could afford a controllable power supply in single-phase homogeneous materials, breaking through the bandgap limitation for typical built-in asymmetry systems (Schottky barriers or p - n junctions).¹¹ Secondly, in contrast to conventional asymmetric systems of Schottky barrier and p - n junction, ferroelectrics don't need complicated interface engineering and fabrication process.¹² Thirdly, spontaneous polarization in ferroelectrics can be modified by electrical field, offering multiple electrically tunable functionalities.¹³ In addition, ferroelectric creates an internal electric field at least one order of magnitude higher than that in the p - n junction.¹⁴

By the way, symmetry-breaking ferroelectric phase transition simultaneously accompanies with the change of pyroelectric currents. In principle, almost all the polar materials might be expected to manifest pyroelectric effects and enable response to thermal change. Under external variable radiation of thermal energy, an electric current will be produced in such material, being proportional to the rate of temperature change.¹⁵ As depicted in Figure 3b in manuscript, our crystal exhibits obvious pyroelectric response during the phase transition. That is, such device has potential applications in temperature sensing.

References

[11] Spanier, J. E. et al. Power conversion efficiency exceeding the Shockley-Queisser limit in a ferroelectric insulator. *Nat. Photonics* **10**, 611-616 (2016).

- [12] Yang, S. Y. et al. Above-bandgap voltages from ferroelectric photovoltaic devices. *Nature Nanotech.* **5**, 143-147 (2010).
- [13] Li, J. K. et al. Self-driven visible-blind photodetector based on ferroelectric perovskite oxides. *Appl. Phys. Lett.*, **110**,142901 (2017).
- [14] Lejman, M. et al. Giant ultrafast photo-induced shear strain in ferroelectric BiFeO₃. *Nat. Commun.* **5**, 4301 (2014).
- [15] Sun, Z. H. et al. Ultrahigh Pyroelectric Figures of Merit Associated with Distinct Bistable Dielectric Phase Transition in a New Molecular Compound: Di- n - Butylammonium Trifluoroacetate. *Adv. Mater.* **27**, 4795-4801 (2015).

Q2. Application for Chiral-optics with chiral ferroelectric material is a motivation in this field (See Guankui Long *et al.*, Chiral-perovskite optoelectronics). Will the material used in the current draft be useful for making chiral ferroelectric and be used as circular-polarization photon detectors? These should be discussed.

Answer: Thanks a lot for the reviewer's useful suggestion. We agree with the reviewer's statement that chiral ferroelectric material is a hot topic in ferroelectric field. According to the reviewer's valuable suggestion, we have consulted and cited the literature review of "Chiral-perovskite optoelectronics" by Guankui Long *et al.* (Please see Ref.14 in revision).¹⁶ In principle, chiral ligands can be used to directly synthesize chiral perovskites. When a perovskite material incorporates chiral organic molecules, they can impart their chiral properties to it. For this paper, we design the two-dimension sandwich-like architecture perovskite ferroelectric making use of the small-size cations (ethylammonium) and bulky alkylammonium cations (isopentylammonium), named **PEPI**. The organic components (ethylammonium and isopentylammonium) not possess chiral structure. Besides, at ferroelectric phase, **PEPI** crystallizes in the orthorhombic system with a achirality space group $Cmc2_1$ (point group $mm2$).

Recently, Dan Li *et al.* report that in addition to the typical chiral point group (1, 2, 3, 4, 6, 222, 422, 432, 32, 622, 23), the point group (m , $mm2$, $\bar{4}$, $\bar{4}2m$) also possess chiroptical activity.¹⁷ Hence, according to the reviewer's suggestion, we fabricated the photodetector device on the single crystals of **PEPI** to check the circular-polarization

photodetection. As shown in the following **Figure 5**, current-voltage curves under LCP-520 nm, RCP-520 nm and unpolarized-520 nm light illumination exhibit no obvious difference. That is, **PEPI** may not be able to differentiate left-handed circularly polarized (LCP) and right-handed circularly polarized (RCP) light. Moreover, structural analysis shows that the flack factor of **PEPI** is 0.44 at ferroelectric phase. Such a flack factor (close to 0.5) also indicates that **PEPI** maybe not suitable used for circular-polarization photodetection.

Figure 5. The current-voltage curves of **PEPI** photodetector device under dark, LCP-520, RCP-520 and unpolarized-520 nm light illumination with the light intensity of 25 mW/cm².

References

- [16] Long, G. K. et al. Chiral- perovskite optoelectronics. *Nature Reviews Materials* **5**, 423-439 (2020).
- [17] Xu, L. L. et al. Chiroptical Activity from an Achiral Biological Metal-Organic Framework. *J. Am. Chem. Soc.* **140**, 11569-11572 (2018).

Q3. The use of "1" as the presentation of the material reads very strange, this should be replaced with other words.

√**Answer:** Thanks a lot for the reviewer's useful suggestion. According to the suggestion, we have replaced "1" with "PEPI" to name our compound (isopentylammonium)₂(ethylammonium)₂Pb₃I₁₀ (Please see the revised manuscript).

Q4. The detection speed seems very fast, but seems beyond author's instrument detection limit. It is good to see the exact response time for such a fast detector.

√**Answer:** Great thanks for the reviewer's positive comments on our manuscript. Responding time is one of the most crucial figure-of-merits for photodetectors and quick responding rate could broaden the application scope of photoelectric devices, including ultrafast dynamic process simulation, high-frequency optical communication, and fast imaging. Here, the response speed of our detector was measured by the transient photocurrent method,¹⁸⁻²⁰ using the 355 nm pulse laser with a width of 3 ns and high-speed oscilloscope (Tektronix MDO3014) with the frequency of 1 GHz to record response signals.

References

[18] Shen, L. et al. A Self-Powered, Sub-nanosecond-Response Solution-Processed Hybrid Perovskite Photodetector for Time-Resolved Photoluminescence-Lifetime Detection. *Adv. Mater.* **28**, 10794-10800 (2016).

[19] Bao, C. X. et al. Low-Noise and Large-Linear-Dynamic-Range Photodetectors Based on Hybrid-Perovskite Thin-Single-Crystals. *Adv. Mater.* **29**, 1703209 (2017).

[20] Wei, H. T. et al. Trap Engineering of CdTe Nanoparticle for High Gain, Fast Response, and Low Noise P3HT: CdTe Nanocomposite Photodetectors. *Adv. Mater.* **27**, 4975-4981 (2015).

Q5. Stability is normally a problem for such detectors. Is there any such test?

√**Answer:** Thanks a lot for the reviewer's useful suggestion. The intrinsic thermal and phase stability of materials is quite essential to the subsequent long-term device performances. Firstly, according to the suggestion, thermogravimetric-differential

scanning calorimetry (TG-DSC) analysis was carried out on a Netzsch STA 449C unit (Please see above **Figure 3**). The TG-DSC curve indicates that **PEPI** possesses a high thermal stability up to 560 K. Secondly, as shown in the following **Figure 6**, the current-voltage curves of the crystal detector after 7 days (under the relative humidity of $40\% \pm 5\%$ at room temperature) exhibit slight difference compared with the fresh device. These above results disclose that **PEPI** possesses fairly good thermal and environmental phase stabilities.

Figure 5. Current-voltage curves of the freshly-prepared detector and the device after 7 days aging in the air under the relative humidity of $40 \pm 5\%$ at room temperature.

Reviewer #1 (Remarks to the Author):

The authors have made very detailed responses to the comments of all reviewers. More importantly, their responses are serious and persuasive, so the manuscript was greatly improved. I think the article is acceptable for publication and does not need further modification.

Reviewer #2 (Remarks to the Author):

The raised points have been well solved. According to this reviewer's judgment, the present work will be an outstanding and exciting contribution to ferroelectricity and related fields. Thus, this reviewer does recommend the publication of the present article in Nature Communications without further changes.

Reviewer #3 (Remarks to the Author):

Authors have implemented additional experiments to address previous comments and reported a good stability of the device. It represents a good breakthrough in this field and I now recommend its publication in Nature communications.

COMMENTS TO AUTHOR:

Reviewer #1:

The authors have made very detailed responses to the comments of all reviewers. More importantly, their responses are serious and persuasive, so the manuscript was greatly improved. I think the article is acceptable for publication and does not need further modification.

√**Answer:** Great thanks for the reviewer's positive comments on our manuscript.

Reviewer #2:

The raised points have been well solved. According to this reviewer's judgment, the present work will be an outstanding and exciting contribution to ferroelectricity and related fields. Thus, this reviewer does recommend the publication of the present article in *Nature Communications* without further changes.

√**Answer:** Thanks a lot for the reviewer's positive comments on our manuscript.

Reviewer #3:

Authors have implemented additional experiments to address previous comments and reported a good stability of the device. It represents a good breakthrough in this field and I now recommend its publication in *Nature communications*.

√**Answer:** Great thanks for the reviewer's positive comments on our manuscript.